# Use of simulation training to teach the ABCDE primary assessment: an observational study in a Dutch University Hospital with a 3–4 months follow-up

Amanda M Drost-de Klerck [ID] ,[1] Tycho J Olgers,[2] Evelien K van de Meeberg,[1] Johanna Schonrock-Adema,[3] Jan C ter Maaten[2]

[1]Emergency Department, University Medical Center Groningen, Groningen, The Netherlands
[2]Department of Internal Medicine, University Medical Center Groningen, Groningen, The Netherlands
[3]Institute for Medical Education, University Medical Center Groningen, Groningen, The Netherlands

**Correspondence to**
Amanda M Drost-de Klerck;
a.drost@umcg.nl

## ABSTRACT

**Objectives** To investigate short-term and long-term effectiveness of simulation training to acquire a structured Airway Breathing Circulation Disability Exposure (ABCDE) approach for medical emergencies; and to examine which skills were learnt and maintained best.

**Design** An observational study with a 3–4 months follow-up.

**Setting** Skills center of the University Medical Center Groningen.

**Participants** Thirty voluntary participants (21 females and 9 males; 27±2.77 years) of a simulation-based course.

**Intervention** A 2-day ABCDE-teaching course for residents and non-residents. The course encompasses 24 simulations in which participants perform primary assessments of acute ill patients. Video recordings were taken of each participant performing a primary assessment, before (T1), directly after (T2) and 3–4 months after the intervention (T3).

**Main outcome measures** Physicians' performance in the ABCDE primary assessment at T1, T2 and T3. Two observers scored the primary assessments, blinded to measurement moment, using an assessment form to evaluate the performance with regard to skills essential for a structured ABCDE approach. The Friedman and Wilcoxon signed-rank test were used to compare physicians' performances on the subsequent measurement moments.

**Results** The mean ranks on the total primary assessment at T1, T2 and T3 were 1.14, 2.62 and 2.24, respectively, and were significantly different, ($p<0.001$).
The mean ranks on the total primary assessment directly after the course (T2 vs T1 $p<0.001$) and 3–4 months after the course (T3 vs T1 $p<0.001$) were significantly better than before the course. Certain skills deteriorated during the follow-up. Strikingly, most skills that decrease over time are Crew Resources Management (CRM) skills.

**Conclusion** A course using simulation training is an effective educational tool to teach physicians the ABCDE primary assessment. Certain CRM skills decrease over time, so we recommend organising refresher courses, simulation team training or another kind of simulation training with a focus on CRM skills.

## Strengths and limitations of this study

► This is an observational study to investigate the short-term and long-term effect of a simulation course.

► This study used the same environment for course and study.

► The observers were blinded to measurement moment during this study on participants' performance on the primary assessment using the Airway Breathing Circulation Disability Exposure approach.

► The fact that the observers might have been also the instructor of some study participants, may have influenced their ratings for a few participants, but potential bias was minimised by offering the videos in random order and blinded to measurement moment, analysis showed a moderate to high interobserver reliability and our study focused on the outcomes at group level and not individual outcomes.

► There might be bias in the fact that we do not know if the participants have prepared for the study scenario at T1 and T3 (the Hawthorne effect), whereas preparation for T3 might result in an overestimation of the actual course effectiveness at follow-up, preparation at T1 might result in an underestimation of the effectiveness of the course on both T2 and T3.

## INTRODUCTION
### Background

In emergency medicine, assessing incoming patients in life-threatening conditions according to a structured approach is considered essential for successful resuscitation. The most widely used structured approach for early recognition and immediate treatment of life-threatening conditions is the 'Airway Breathing Circulation Disability Exposure' (ABCDE) approach. The ABCDE approach is taught in the Advanced Trauma Life Support since 1978 and has been the standard approach in trauma since.[1–3]

The use of the ABCDE primary assessment has also increased in other medical emergencies in the recent years.[4–7]

Using the ABCDE approach likely improves outcomes by helping healthcare professionals focus on the most life-threatening clinical problems and perform immediate resuscitation. Although solid empirical evidence for the usefulness of the ABCDE approach and its clinical benefits to patients is limited,[1 2] the importance of early treatment has been recognised in several emergencies such as trauma, stroke, sepsis and shock.[1–5 8–11]

## Importance

The Dutch inspection for healthcare requires that physicians treating non-trauma patients in the emergency department (ED) are ABCDE qualified.[12] Therefore, completing an ABCDE course is mandatory for physicians who work at the ED. These courses usually contain lectures and simulation scenario training. Despite the wide use of simulation training for teaching the systematic ABCDE approach, little research has been done to analyse the effectiveness of simulation training in acquiring this structural approach. Simulation training has been proven to be effective for learning technical skills and maintaining skills that are not frequently used in daily practice, like airway management and surgical skills.[13–15] Simulation training can also improve communication, efficiency and safety during teamwork.[16–18] A few studies based on self-perceptions showed that simulation training improved participants' confidence levels; they felt more competent in applying the ABCDE approach and several other skills.[3 19–21]

To our knowledge, it has not been investigated before whether simulation training actually improves physicians' skills in performing the structured ABCDE approach.

Our study focused on the effectiveness of simulation training to acquire a structured ABCDE approach. Our main goal was to analyse the short-term and long-term effectiveness of simulation training to acquire a structured ABCDE approach. We analysed the improvement in physicians' primary assessment scores as a result of the ABCDE simulation training.

We also investigated whether the skills acquired were maintained over a period of 3–4 months and which skills and competences were learnt and maintained best.

## METHODS
### Study design

We conducted an observational study to investigate short-term and long-term effectiveness of a 2-day simulation-based ABCDE teaching course. The measurements through video recordings were obtained before (T1), directly after (T2) and 3–4 months after the intervention (T3).

Three simulation scenarios (A, B and C) with different medical emergencies were specifically designed for this study. Scenario A was a case with pneumosepsis and hypoglycaemia, a partially obstructed airway due to low consciousness and shock. Scenario B concerned a case with obstructive shock caused by pulmonary embolism and an opioid overdose with altered consciousness. Scenario C was a case with meningococcal sepsis with a partially obstructed airway due to low consciousness, bronchospasm and shock. We have designed three different and realistic scenarios with comparable difficulty by creating a life-threatening condition which needs resuscitation in three of the five main items from the ABCDE.

To prevent bias caused by the type or difficulty of the simulation, we varied the order in which participants had to complete the three simulation scenarios in such a way that the different scenario sequences were equally divided over T1, T2 and T3 (participant 1: T1 scenario A, T2 scenario B, T3 scenario C; participant 2: T1 scenario B, T2 scenario C, T3 scenario A; participant 3: T1 scenario C, T2 scenario A, T3 scenario B, etc). We made a schedule in which the order of the scenarios was prescribed for each participant and participants were divided over the schedule in order of inclusion.

We developed an assessment form (figure 1) to evaluate the participants' performance regarding skills and competences essential to assess medical emergencies. The assessment form was divided in six categories; five concerned the ABCDE structure and the sixth contained remaining items. The remaining items focus on some Crew Resources Management (CRM) skills, like collaboration, communication, acknowledge own boundaries, and leadership. In each category, the skills or competences could be rated on a two-point (agree, not agree, does not apply) or four-point scale (agree, partially agree, partially not agree, not agree or does not apply). We have added the option 'does not apply', because some skills were not required in some simulation scenarios. In the categories B, C, D and E the number of examined items during the physical examination were also scored. The following items could be scored; in the B: skin colour, trachea position, respiratory rate, thorax excursions, breathing effort, lung percussion, lung auscultation and saturation; in the C: circulation of extremities, central pulse, heart rate, blood pressure, capillary refill, central venous pressure, heart sounds; in the D: Glasgow Coma Scale, pupils, neck stiffness, glucose; in the E: temperature, head to toe examination (figure 1).

### Intervention

The ABCDE course is a 2-day course for non-residents and first year residents which exists for 10 years now. For most participants, it was a mandatory course that they need to pass before they were allowed to work in the ED. The course consisted mainly of simulation training and two theoretical lectures about airway management and Advanced Life Support (ALS). Previous to this course, the participants received a book with chapters describing the ABCDE approach and various acute medical emergencies.

The course focused on learning to recognise and treat life-threatening conditions, but also paid attention

**Primary assessment**

| | Item | Rating |
|---|---|---|
| **A** | - examines the airway | agree / disagree |
| | - mentions obstructed airway | agree / disagree/ d.n.a. |
| | - applies airway maneuvers | agree / partially agree / partially disagree / disagree / d.n.a. |
| | - applies oxygen | agree / disagree/ d.n.a. |
| **B** | - examines B completely (color, trachea, resp. rate, excursions, breathing effort, percussion, auscultation, saturation) | 1 - 2 - 3 - 4 - 5 - 6 - 7 – 8 |
| | - gives nurse the right orders | agree / partially agree / partially disagree / disagree / d.n.a. |
| | - mentions abnormal findings | agree / partially agree / partially disagree / disagree / d.n.a. |
| | - recognizes life-threatening conditions | agree / disagree / d.n.a. |
| | - orders right additional diagnostic tests | agree / disagree / d.n.a. |
| | - mentions conclusions/interpretation | agree / partially agree / partially disagree / disagree |
| | - resuscitates adequately | agree / partially agree / partially disagree / disagree / d.n.a. |
| **C** | - examines C completely (circulation of extremities, central pulse, heart rate, blood pressure, cap.refill, CVP, heart sounds) | 1 - 2 - 3 - 4 - 5 - 6 - 7 |
| | - gives nurse the right orders | agree / partially agree / partially disagree / disagree / d.n.a. |
| | - mentions abnormal findings | agree / partially agree / partially disagree / disagree / d.n.a. |
| | - recognizes life-threatening conditions | agree / disagree / d.n.a. |
| | - orders right additional diagnostic tests | agree / disagree / d.n.a. |
| | - mentions conclusions/interpretation | agree / partially agree / partially disagree / disagree |
| | - resuscitates adequately | agree / partially agree / partially disagree / disagree / d.n.a. |
| **D** | - examines D completely (Glasgow Coma Scale, pupils, neck stiffness, glucose) | 1 - 2 - 3 – 4 |
| | - applies EMC correctly | agree / partially agree / partially disagree / disagree |
| | - mentions abnormal findings | agree / partially agree / partially disagree / disagree/ d.n.a. |
| | - recognizes life-threatening conditions | agree / disagree / d.n.a. |
| | - orders right additional diagnostic tests | agree / disagree / d.n.a. |
| | - mentions conclusions/interpretation | agree / partially agree / partially disagree / disagree |
| | - resuscitates adequately | agree / partially agree / partially disagree / disagree / d.n.a. |
| **E** | - examines E completely (temperature, head to toe) | 1 – 2 |
| | - gives nurse the right orders | agree / partially agree / partially disagree / disagree/ d.n.a. |
| | - mentions abnormal findings | agree / partially agree / partially disagree / disagree/ d.n.a. |
| | - orders right additional diagnostic tests | agree / disagree / d.n.a. |
| | - mentions conclusions/interpretation | agree / partially agree / partially disagree / disagree |
| | - resuscitates adequately | agree / partially agree / partially disagree / disagree / d.n.a. |
| **R** | - asks for help adequately | agree / partially agree / partially disagree / disagree |
| **E** | - communicates clearly | agree / partially agree / partially disagree / disagree |
| **M** | - summarizes adequately | agree / partially agree / partially disagree / disagree |
| **A** | - draws the right conclusions | agree / partially agree / partially disagree / disagree |
| **I** | - clinical reasoning is adequate | agree / partially agree / partially disagree / disagree |
| **N** | - works structured | agree / partially agree / partially disagree / disagree |
| **I** | - stays calm | agree / partially agree / partially disagree / disagree |
| **N** | - shows confidence | agree / partially agree / partially disagree / disagree |
| **G** | - shows good leadership | agree / partially agree / partially disagree / disagree |

**Figure 1** Assessment form used by the observers. CVP, Central Venous Pressure.

to some CRM-skills necessary for an efficient ABCDE approach.

This course was given in the skills center in a room similar to a resuscitation room in the ED. The patient simulator used was a Laerdal Resusci Anne SkillTrainer with an upgrade Vitale Signs Sim Software Complete package. This simulator features heart and lung sounds, chest excursions, pulse and can show all vital signs on a separate monitor. With a separate computer, the sounds and vital signs can be changed during the scenario, to simulate several acute medical conditions.

Each course group consisted of six participants and two instructors. During the simulation rounds the group was split in half and two scenarios were run simultaneously in two separate rooms.

The course encompassed a total of 24 simulations with a patient simulator in which participants perform the primary assessment of acute ill patients. In each scenario, the role of physician, 'non-obstructive nurse'

and observer were assigned to the three participants. One of the instructors operated the simulator and led the debriefing afterwards.

In eight scenarios, the participants fulfilled the role of physician; in the other scenarios, they carried out the role of 'non-obstructing nurse' or observer.

The participants received a certificate if they passed the theoretical test and if they were, according to the instructors, capable of performing a structured primary assessment of an acute ill patient, with recognition and resuscitation of life-threatening conditions and adequate CRM skills.

All course instructors have to follow a formalised educational programme to become an instructor: First they have to pass the course as participant and have to work in the field of emergency medicine or acute care. Second they need to follow a 2-day generic instructor course specifically developed for simulation training. Then they have to act as assistant trainer for at least two courses and

they need to write a report reflecting on their own role as instructor. Finally, they are observed by an experienced instructor to become certified. As instructor, they have to teach the course least twice a year to stay competent and they need to follow the course-specific instructors day each year.

### Study setting and population

This study was conducted in the same skills center as were the course took place. During the video recordings, the simulator, materials and environment were also the same as during the course.

We approached all participants prior to this 2-day course by email and invited them to participate in the study between August 2012 and December 2013.

We endeavoured to achieve a save response environment by a statement in the invitation email that declining to participate in the study would not influence their course results. All participants participated voluntarily, they knew all information about the investigation and they could withdraw from the study at any moment, all provided verbal consent.

The three measurement moments were scheduled in consultation with the participants, separate from the course. For each measurement moment, study participants were instructed to act in a simulation scenario as physician and to perform a primary assessment according to the ABCDE approach. One of the researchers participated as 'non-obstructive' nurse and one researcher operated the simulator and computer.

### Patient and public involvement

Participants of this study were not involved in the development of the research question, design or outcome measures. Some participants of the study encouraged others to participate, but they were all voluntarily included. The results of the study will be available for the participants on request.

### Study protocol

The first recording (T1) took place 1–2 weeks prior to the course. The second recording (T2) took place within 1 week after the course. The third recording (T3) took place between 3 and 4 months after the course.

The research team consisted of five physicians, who were also course instructors. They were all instructed in detail to only facilitate the simulation and not help the participant in any way.

The observers were two emergency physicians, who were also course instructors, but who were not part of the research team and therefore not involved in the recordings. The observers received specific instructions how to score each item on the assessment form. They independently rated the recorded primary assessments in random order and were blinded to the measurement moment.

### Measurements

Each skill or competence on the assessment form had a lowest score of 0 and a highest score of 1, so the weight of each item was the same, independent of the two-point (0=not agree, 1=agree, not applicable=missing value) or four-point scale (agree=1, partially agree=0.67, partially not agree=0.33, not agree=0, not applicable=missing value). This was the same for the number of examined items during the physical examination. For example in the B there was a maximum of eight items to examine during physical examination. If one item was examined the score was $1/8 = 0.125$, if two items were examined the score was $2/8 = 0.25$, if three items were examined, the score was $3/8 = 0.375$. So, the highest possible score on complete examination in the B was $8/8 = 1$.

Because some skills or competences were marked as not applicable, we calculated mean scores in each category (A, B, C, D, E and remaining items) based on the skills and competences which actually were applicable. In each category, the maximal score to obtain was 1. Therefore, the maximal total score to obtain on the primary assessment for each scenario was 6 and the minimal score was 0.

### Data analysis

To perform the statistical analysis, IBM SPPSS V.23.0 was used. In all analyses, a $p < 0.05$ was regarded as significant.

The interobserver reliability between the scores given by the two observers for the three different time measurements was calculated using the Spearman's rank correlation test for T1, T2 and T3 (resp. R 0.81, 0.61 and 0.80). This interobserver reliability was generally high enough to average the scores of the two observers for use in further analyses, as a correlation coefficient lower than 0.5 is considered as weak correlation, a correlation coefficient between 0.5 and 0.7 is considered as moderate correlation, a correlation coefficient between 0.7 and 0.9 is considered as high correlation and a correlation coefficient between 0.9 and 1 is considered as very high correlation.

We used the Friedman test for three related samples to analyse whether the total primary assessment scores of the entire group of participants differed between the three measurement moments. The Friedman test calculates and compares the mean ranks at T1, T2 and T3. The mean rank is calculated on a scale from 1 to 3, because three measurements are ranked, 1 is the best rank and 3 the worst. The mean rank is calculated by ranking the score of each participant on T1, T2 and T3 and then calculating the mean rank of the entire group on T1, T2 and T3.

We used the Wilcoxon signed-rank test for two related samples to analyse whether the total primary assessment scores of the entire group of participants differed between two measurement moments and whether each skill or competence differed between two measurement moments. The Wilcoxon signed-rank test also uses the mean ranks.

Finally, we applied the Holm correction to reduce the possibility of getting a statistically significant result (type I error) when performing multiple tests.

| Time | N | Median | 25th percentile | 75th percentile | Mean rank | Wilcoxon signed-rank test |
|---|---|---|---|---|---|---|
| 1 | 30 | 2.79 | 2.31 | 3.53 | 1.14 | T1 <T2, p<0.001 |
| 2 | 30 | 5.22 | 4.57 | 5.43 | 2.62 | T2 >T3, p<0.05 |
| 3 | 29 | 4.70 | 4.20 | 5.30 | 2.24 | T3 >T1, p<0.001 |

**Table 1** Scores on the total primary assessment of the whole group at T1, T2 and T3

## RESULTS

### Characteristics of study subjects

Between August 2012 and December 2013, 27 courses were given to six participants each. From the total of 162 course participants 30 participants volunteered for this study. Twenty-one were female, nine were male. Their mean age was 27 years (range 24–35, SD 2.77), their mean work experience was 11 months (range 0–48, SD 14.4). Most participants did not have any experience with simulation training at all (18 out of 30), some participants had done some training in their own department, like ALS or Basic Life Support (BLS) (7 out of 30), from five participants we do not know whether they had any experience with simulation training.

The video recording of T3 of one participant was lost due to technical problems.

### Main results

The median total score on the primary assessment was 2,79 at T1, 5.22 at T2 and 4.70 at T3 (table 1 and figure 2).

The mean ranks of the entire group on the total primary assessments at T1, T2 en T3 were 1.14, 2.62 and 2.24, respectively (table 1), and they were significantly different, (p<0.001).

The mean rank on the total primary assessment at T2 (directly after the course) was significantly higher than the mean rank at T1 (before the course, table 1). The mean rank on the total primary assessment at T3 (3–4 months after the course) was significantly lower than the

mean rank at T2, but remained significantly higher than the mean rank at T1 (table 1).

The mean ranks of the separate skills or competences were almost all significantly higher at T2 than at T1 (34 out of 40). With respect to the remaining skills, four could not be included in our analyses as they were scored too often as 'does not apply', which rendered the number of observations for those skills <N=10, which was too low to ascertain differences in a reliable way. For only three skills—'examines the airway', 'orders additional diagnostics in the B' and 'resuscitates adequately in the E'—we did not find a significant difference between T1 and T2.

Most of the separate skills did not show significant differences between mean rank at T2 and T3 (30 out of 40). Some skills (7 out of 40) had a significantly lower mean rank at T3 than at T2, but significantly higher than at T1 (table 2).

## DISCUSSION

Our study is the first to show the effectiveness of a course using simulation to teach physicians the ABCDE approach for the assessment of medically ill patients. We found that the positive effect on performing a primary assessment according the ABCDE approach persisted even 3–4 months after completing the course.

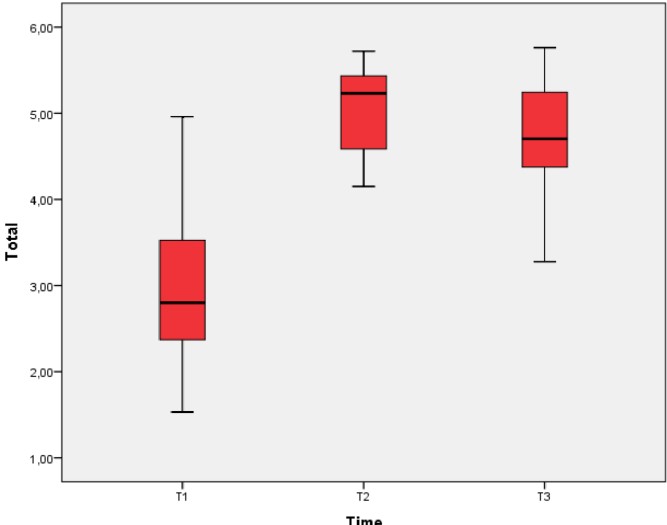

**Figure 2** Boxplot showing median and IQR of total score on primary assessment at T1, T2 and T3.

**Table 2** Outcomes for separate skills and competences that decreased between T2 and T3

| Skill/competence | Wilcoxon signed-rank test | | |
|---|---|---|---|
| | N= | | |
| Mentions abnormal findings in the C | 29 | T2 >T3<br>T1 <T3 | p<0.01<br>p<0.01 |
| Recognises life-threatening conditions in the C | 28 | T2 >T3<br>T1 <T3 | p<0.05<br>p<0.01 |
| Mentions conclusions in the C | 28 | T2 >T3<br>T1 <T3 | p<0.05<br>p<0.01 |
| Examines the D completely | 29 | T2 >T3<br>T1 <T3 | p<0.01<br>p<0.001 |
| Communicates clearly | 29 | T2 >T3<br>T1 <T3 | p<0.05<br>p<0.001 |
| Shows confidence | 29 | T2 >T3<br>T1 <T3 | p<0.05<br>p<0.001 |
| Shows good leadership | 29 | T2 >T3<br>T1 <T3 | p<0.05<br>p<0.001 |

Our findings corroborate outcomes of other studies showing that simulation training in health professions education was consistently associated with large effects on knowledge, skills and behaviour.[22 23] Our findings are also in line with previous research showing that simulation training establishes, corrects, and confirms knowledge and skills of the ABCDE approach and afterwards participants felt more competent in applying the ABCDE approach.[3 19–21]

In the follow-up, we found decline in participant performance on some skills of the primary assessment. Strikingly, most skills that decrease over time are CRM skills (table 2). This is illustrated by a decrease in time of 'recognition of life-threatening conditions in the C', while the scores on the resuscitation skills did not decline. It is possible that this lower score reflects 'not thinking out loud' rather than failing to recognise a life-threatening condition.

This decrease in CRM skills suggests that focusing on CRM skills by refresher courses or team training, after completing a simulation course, may be an important topic for physicians to maintain their skills. The positive effect of team training for these non-technical skills has already been shown.[16–18]

Another skill that does not yield scores as high as most other skills after 3–4 months is a complete examination of the Disability. A possible explanation for this finding is that the participants decide on the level of consciousness of the patient, determined by the Glasgow Coma Scale, whether it is necessary to examine certain components of the Disability. The performance of the Glasgow Coma Scale does not decrease over time. This finding is in line with previous research from our group on primary assessment completeness showing that during the primary assessment in the ED, residents and experienced staff have equal, but not maximum ABCDE completeness scores (83 instead of 100).[7] Fernández-Méndez et al also showed that professional lifeguards failed to fully perform the ABCDE sequence and spend more time in the Circulation step, because they spent more time in steps considered most important.[5]

These outcomes may reflect that a score of 100 on the ABCDE approach is not necessary to exclude potential life-threatening diseases or stabilise the patients.

## Limitations

It is not possible to define the impact of the book and the lectures that are also part of the course, on the measured improvements in performance on the primary assessment. Outcomes from the regular course evaluation—not part of this study—indicated that the simulation training was the most powerful educational tool and accounted for most of the improvements. This feedback is in line with previous research showing that adding simulation training to a curriculum with lectures of medical students is associated with higher oral exam scores and higher overall course grades.[22]

This study evaluated a course with instructors who are experts in the field of acute medicine, and experienced and certified course instructors. It is known that simulation-based education is most effective if guided by a safe and efficient debriefing and that debriefing can be challenging.[23 24] We do not know if simulation training with debriefing by less experienced instructors may have less effect.

During the study, we have deliberately chosen for a researcher participating as 'non-obstructive nurse' in the measurement to minimise potential bias caused by help from the 'non-obstructive nurse'. The researcher knew the research questions and was instructed in detail to only follow instructions from the participant and not help in any way.

We did not schedule the researchers and operators with an equal distribution over the measurement moments, but all five researchers rotated between roles of the nurse and operator on own initiative. We, therefore, think that the bias of the non-obstructive nurse influencing the participant is negligible.

Another limitation of this study is that it was not possible to assess all specific skills in each simulation scenario, because they were often scored as 'does not apply'. The amount of not applicable rated items was between 0 and 3 in 10 items, between 3 and 10 in 5 items, between 10 and 20 in 2 items and in 4 items the not applicable rated items was >20. This limitation probably did not influence the results because items often scored as 'does not apply' do not impact discriminating in quality of performance.

The sample size was chosen without power analysis, because we did not know the expected effect. This relative small sample size of 30 participants already showed large significant differences. In our statistical analysis, we accounted for a small sample size by using the Wilcoxon signed-rank test.

The observers may have been the instructor during the course of some study participants and we cannot exclude that this may have influenced their ratings for some of them. This potential bias was minimised by offering the videos in random order and blinding the observers to the measurement moment. Also, our study focused on the outcomes at group level and not individual outcomes and the interobserver analysis showed a moderate to high interobserver reliability.

The measurement moment of T3 varied between 3 and 4 months after T2. We do not know if this range of 1 month between T2 and T3 have caused a variation in performance at T3.

Some participants had experience with simulation training. These participants might have had a higher score on T1 what might have caused an underestimation of the difference between T1 and T2.

Finally, the participants knew they were a study subject and when the recordings were scheduled. We do not know if they prepared for the study scenarios. Modifying behaviour in response to the awareness of being observed is known as the Hawthorne effect. This may

have influenced the study scores at T1 and T3, whereas preparation for T3 might result in an overestimation of the actual course effectiveness at follow-up, preparation at T1 might result in an underestimation of the effectiveness of the course on both T2 and T3.

## CONCLUSION

A course with simulation training is an effective educational tool to teach physicians to perform a structured primary assessment using the ABCDE. This competence is largely remained after 3 to 4 months. CRM skills tend to decrease over time, so we recommend organising refresher courses, simulation team training or another kind of simulation training with a focus on CRM-skills.

**Acknowledgements** We want to thank Kinge van der Heide for her help with the recordings. We want to thank Mirjam Doff-Holman and Martine Oosterloo for the assessment of all recordings.

**Contributors** AMD-dK and JCtM have conceived the study and created the assessment form. AMD-dK, TO, EvdM and JCtM collected the data. AMD-dK is guarantor and did the literature search, managed the data and performed data analysis and drafted the manuscript. JS-A advised regarding to the data analysis. All authors contributed to the final manuscript, revision of the manuscript and final approval of the manuscript. The corresponding author attests that all listed authors meet authorship criteria and that no others meeting the criteria have been omitted.

**Funding** This study was only funded by the Emergency Department of the University Medical Center Groningen.

**Competing interests** None declared.

**Patient consent for publication** Not required.

**Ethics approval** Ethical approval was waived by our medical ethics committee (METc UMC Groningen) as this research is educational research. We have received an independent review board declaration from our medical ethics committee, which declares that this study fulfils all the requirements for patient anonymity and is in agreement with regulations of our University Hospital.

**Provenance and peer review** Not commissioned; externally peer reviewed.

**Data availability statement** Data are available on reasonable request. Unpublished statistical data will be available on request to the corresponding author.

**ORCID iD**
Amanda M Drost-de Klerck http://orcid.org/0000-0002-5432-7733

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
