## [Reviewer comments · BMJ Open]

ARTICLE DETAILS

TITLE (PROVISIONAL)	Use of simulation training to teach the ABCDE primary assessment: an observational study in a Dutch University Hospital with a 3-4 months follow up.
AUTHORS	Drost- de Klerck, Amanda; Olgers, Tycho; van de Meeberg, Evelien; Schonrock-Adema, Johanna; ter Maaten, Jan

VERSION 1 - REVIEW

REVIEWER	Cristian Abelairas-Gómez Universidade de Santiago de Compostela, Spain
REVIEW RETURNED	17-Jun-2019

GENERAL COMMENTS	Authors present an interesting study about the effect of simulation training in the practical skills of physicians regarding to ABCDE approach. Methodologically, it seems that the manuscript is well designed, but I propose some recommendations in order to make it clearer. General comments: -The instrument described in Figure 1 that was used to evaluate the participants should be described much better.-The different scales, the punctuations of each scenario in each test are aspects that are not very well explained.-In the title should be placed that it is an observational study and show de 3-months follow-up (also in the objective of the introduction and abstract).-P values should not be shown as “p = 0.000”; please, replace by “p < 0.001”.-All abbreviations should be explained the first time that appear in the text with the abbreviation in brackets. Specific comments: Abstract: -What is “total rank score” should be explained. Methods -The manuscript must state that participants participated voluntary and all the requirements for this kind of research.-The method in order to randomize the order of the simulations should be described.-Please, describe more information about “to offer variable and realistic scenarios with the same degree of difficulty”. How was “difficulty” measured?
---

-Figure 1:
 +The explanation of Figure 1 do not correspond with Figure 1.
 +It is described in the text that "In each category, the skills of competences could be rated on a 2 (agree, not agree) or 5-point scale (agree, partially agree, partially not agree, not agree or does not apply)". However, in the Figure 1 there are categories with 3 and 4-point scale.
 +Why does not airway assessment have the category "examines A completely" ?
 +Please, describe the number-scale in each section of ABCDE approach.
 +Authors state that "the assessment form was divided in 5 categories...". However, 6 categories are described in Figure 1.
 +What is exactly the point scale "does not apply"? Does it means that participant did not the step of the ABCDE approach? Or is it possible that several steps were not necessary in some simulation scenarios? This information should be explained more accurate.
 +In general, the instrument should be explained in much more detail. Each assessment include general information about orders, deviating findings, interpretations... but it is not described how authors quantified more specific practical skills as breathing assessment, thoracic symmetry, pulse, assessment of neurological status...

-Were participants asked for previous training in the ABCDE approach? All possible information about previous training and knowledge of the participants in this regards should be described.
 -How did the instructors of the ABCDE course know that participants were "capable of performing a structured primary assessment"? Did participants pass only a theoretical test or both, theoretical and practical?
 -Please, add the duration of the course (theoretical & practical) and ratio instructors:pupils.
 -Authors state that "the third and last recording (T3) was taken approximately three months after the course". I understand the difficult to minimize the bias and get the same time for all the participants, but could you provide the time of the shorter and larger follow-up?
 -Information of the section "Measurements" can be placed in other sections of the manuscripts.
 -The way in how the skills and competences were quantitatively measured should not be placed in Data analysis. In the section "Measurements" could place it with a better explanation.
 -Authors state that they used Wilcoxon signed rank test in order to compare each pair of test (T1 vs. T2, T1 vs. T3 & T2 vs. T3). Have you used some correction (i. e. Bonferroni) to minimize Error Type I?

Results

-Please, specify if participants were residents or they had finished the residence, since the course was also for residents.
 -Authors compared different simulation scenarios (3) according with the sum of different point scale. If a participant carried out perfectly each scenario, would he/she reach the same punctuation? This is important since if (for example) the second simulation had a maximum punctuation of 30 points, and the third a maximum punctuation of 25, it would be more difficult to reach better results in T3. This should be explained.
 -Data of Table 1 and Table 2 should be in the same table.
 -I consider that the information of Table 3 is not enough relevant to be in a Table. These skills and competences could not probably be

	analyzed in the type of scenario selected, and maybe they could in another one. Therefore, the low numbers observed by the authors were not necessarily due to participants' performance. -Table 4: Please, add the descriptive statistics of each variable. Discussion -BMJ Open, in the author guidelines, recommend writing short discussions. However, the discussed results are very poor with only four studies referenced. Others studies about ABCDE approach simulation training (Abelsson et al. Learning High-Energy Trauma Care Through Simulation. Clinical Simulation in Nursing 2018) or evaluation (Fernandez-Mendez et al. ABCDE approach to victims by lifeguards: How do they manage a critical patient? A cross sectional simulation study. PLoS One. 2019) could be discussed. -The benefits and challenges of the simulation training should also be discussed. Some references that could help: Cook et al. Technology-enhanced simulation for health professions education: a systematic review and meta-analysis JAMA 2011 / Grant et al. Difficult debriefing situations: A toolbox for simulation educators. Med Teach 2018).
--	---

REVIEWER	Christian Berger Charité Universitätsmedizin Berlin, Germany
REVIEW RETURNED	26-Jun-2019

GENERAL COMMENTS	Manuscript ID: bmjopen-2019-032023 To the editor: I would like to thank you for the opportunity to review the article concerning the short and long term effects of simulated training in teaching the ABCD primary assessment by Drost-de Klerck and colleagues. This work mentioned a relevant subject in the field of medical training, with some methodical and contentual points to address, so it needs review. To the author: In the introduction, you describe the relevance of early resuscitation within the “golden hour”. Further you state, that the ABCDE approach is the primary assessment for trauma since decades. Please provide data regarding this statement and clarify the context of the two statements. In the methods, you mention three simulation scenarios for the measurements. Please present more information about the exact type and content of these scenarios. Where took they place, what kind of equipment and environment was provided for the participants. How many people in which role participate in each scenario in total? Please revise the methodological section and state more clearly, which test was when more used. In this context, please make clearer, why the evaluation of the differences between time points was altered between separate skills (supplemental file). The data collection took place at three different moments, separate from the course. When and how were they scheduled specifically and communicated with the participants. Were they able to prepare before data collection? If observers were identical with the research team and furthermore involved as tutors in the course, please
---

	comment on potential bias. Your study was conducted within the courses between August 2012 and March 2014. How many physicians participate in your courses in total? How was the number of 30 participants in your study selected? Did you perform a power analysis? Has every one of the study participants fulfilled the data collection completely at all three time points? Providing demographical data from your general course population would help the reader. You stated that ethics approval was not required, because intervention was not on behalf of the study due to course participation. What about the T3 measurement, which took place after the course? Please state this. Further, it was possible to not pass the course and it's not clear mentioned, if the observers were involved in data collection or participated as course instructors. Who asked the participants to join the study and when? Were observers or course instructors involved? Asking for a "voluntarily" participation by potential future examiners may influence the participant's decision. Please comment on this. Please state, if the study is enlisted in registry e.g. "clingov". If not so, this should be done. Some of the measured skills were not analyzed as they were scored too often as "does not apply". Please provide information about your cut off for analyzation and how it was calculated. Further, please provide the number of data sets in each category, which were finally analyzed. The discussion sets a focus on communication and leadership. This can be summarized under the topic "CRM". Regarding to your heading and introduction, I recommend focusing on the ABCDE approach or please make clear, how teaching your specific ABCDE course influences the CRM-abilities of the participants. Otherwise, I would suggest changing the topic of your paper. You conclude that simulation training leads to the results of your measurement. Please clarify, why the simulation training and not for example the lecture of the book leads to the rise in the participants performance. Further, is it possible to exclude, that training or experience outside the course leads to the improvement of participants behavior during data collection? Was this evaluated? Pleas discuss this and accordingly adapt the conclusion. Some minor points:  - Please introduce the abbreviation ED in page 7 line 45 before use - For data analysis you state a 5 point scale but using a 4 point scale with the additional option "not applicable", please make this more clear - Page 12 line 35, think you mean "lost" instead "last"
--	---

VERSION 1 – AUTHOR RESPONSE

Reviewer: 1

Reviewer Name: Cristian Abelairas-Gómez

Institution and Country: Universidade de Santiago de Compostela, Spain

Please state any competing interests or state 'None declared': None declared

General comments:

-The instrument described in Figure 1 that was used to evaluate the participants should be described much better.

We have added more detailed information about how Figure 1, the assessment form, was used in a supplemental file for the reviewers. We also added in the methods / study protocol section that the observers received specific information about how to use the assessment form.

-The different scales, the punctuations of each scenario in each test are aspects that are not very well explained.

We have added more detailed information about how Figure 1, the assessment form, was used in a supplemental file for the reviewers. We also added more detailed information in the methods / study design section.

-In the title should be placed that it is an observational study and show de 3-months follow-up (also in the objective of the introduction and abstract).

Thank you for this suggestion to improve our Title. We have changed the title in: Use of simulation training to teach the ABCDE primary assessment: an observational study in a Dutch University Hospital with a 3-months follow up.

-P values should not be shown as “p = 0.000”; please, replace by “p < 0.001”.

We have changed this in the tables in the article and in the supplemental file.

-All abbreviations should be explained the first time that appear in the text with the abbreviation in brackets.

We have explained the abbreviations.

Specific comments:

Abstract:

-What is “total rank score” should be explained.

Thank you for this comment, this helped us to make the data analysis section clearer. We have rewritten the data analysis section and removed this term. We meant the mean rank score on the total primary assessment. We have also adapted the abstract.

Methods

-The manuscript must state that participants participated voluntary and all the requirements for this kind of research.

We strived for a save environment by a statement in the invitation e-mail that declining to participate in the study will not influence the course results.

We added a comment in the methods/study setting and population section.

-The method in order to randomize the order of the simulations should be described.

We have added more detailed information in the methods / study design section.

-Please, describe more information about “to offer variable and realistic scenarios with the same degree of difficulty”. How was “difficulty” measured?

We have described the scenario's in more detail and rewritten the methods / study design section

-Figure 1:

+The explanation of Figure 1 do not correspond with Figure 1.

We have adapted the explanation of Figure 1.

+It is described in the text that “In each category, the skills of competences could be rated on a 2 (agree, not agree) or 5-point scale (agree, partially agree, partially not agree, not agree or does not apply)”. However, in the Figure 1 there are categories with 3 and 4-point scale. We have adapted this in the methods / study design section and refer to Figure 1.

+Why does not airway assessment have the category “examines A completely”?

We have instructed the observers to score the item “examines the airway” as follows:

o Agree =examines the airway completely (looks in the mouth and listens/asks for signs of airway obstruction).

o Disagree = does not examine the airway.

We added a supplemental file for the reviewers.

+Please, describe the number-scale in each section of ABCDE approach.

We have adapted this in the methods / study design section.

+Authors state that “the assessment form was divided in 5 categories...”. However, 6 categories are described in Figure 1. Thank you for noting this inconsistency. We have adapted this.

+What is exactly the point scale “does not apply”? Does it means that participant did not the step of the ABCDE approach? Or is it possible that several steps were not necessary in some simulation scenarios? This information should be explained more accurate.

It means that that certain steps were not required in some simulation scenarios.

We have added this explanation in the methods / study design section.

+In general, the instrument should be explained in much more detail. Each assessment include general information about orders, deviating findings, interpretations... but it is not described how authors quantified more specific practical skills as breathing assessment, thoracic symmetry, pulse, assessment of neurological status...

Thank you for this suggestion to make the methods used clearer. We have added more detailed information in the method/ study design section. And we have added a supplemental file for the reviewers with the instruction to the observers, which explains how the assessment form, Figure 1, was used.

-Were participants asked for previous training in the ABCDE approach? All possible information about previous training and knowledge of the participants in this regards should be described.

We have added this information in the study results / characteristics of the study subjects.

-How did the instructors of the ABCDE course know that participants were “capable of performing a structured primary assessment”? Did participants pass only a theoretical test or both, theoretical and practical?

The participants have to pass a theoretical test and both instructors must agree that the participant has shown enough practical progression and is capable of performing a structured primary assessment and recognizing and treatment of life-threatening conditions. The instructors are experts in the field of acute medicine, and certified as course instructor and trained each year to make a uniform judgment after consulting each other.

-Please, add the duration of the course (theoretical & practical) and ratio instructors:pupils. Thank you for this suggestion. We have added more information in the methods / intervention section.

-Authors state that “the third and last recording (T3) was taken approximately three months after the course”. I understand the difficult to minimize the bias and get the same time for all the participants, but could you provide the time of the shorter and larger follow-up?

The follow-up of all participants followed after three to four months. We have added this information in the methods / study protocol section.

-Information of the section "Measurements" can be placed in other sections of the manuscripts. We have replaced this information in the introduction and we have rewritten the final part of the introduction.

-The way in how the skills and competences were quantitatively measured should not be placed in Data analysis. In the section "Measurements" could place it with a better explanation. We have changed the order/layout of this section and explained it in more detail.

-Authors state that they used Wilcoxon signed rank test in order to compare each pair of test (T1 vs. T2, T1 vs. T3 & T2 vs. T3). Have you used some correction (i. e. Bonferroni) to minimize Error Type I? No we did not use Bonferroni. We did used the Wilcoxon signed-rank test suitable for small sample and not normally distributed population.

Results

-Please, specify if participants were residents or they had finished the residence, since the course was also for residents.

All participants were physicians, some were interns (not in training yet) and some were first year trainee/resident for a specialty. None had finished the residence (specialty training). We have added this information in the methods / study design section.

-Authors compared different simulation scenarios (3) according with the sum of different point scale. If a participant carried out perfectly each scenario, would he/she reach the same punctuation? This is important since if (for example) the second simulation had a maximum punctuation of 30 points, and the third a maximum punctuation of 25, it would be more difficult to reach better results in T3. This should be explained.

We have explained this in more detail in the methods / measurements section.

-Data of Table 1 and Table 2 should be in the same table. We have changed this.

-I consider that the information of Table 3 is not enough relevant to be in a Table. These skills and competences could not probably be analyzed in the type of scenario selected, and maybe they could in another one. Therefore, the low numbers observed by the authors were not necessarily due to participants' performance.

We have removed the table

-Table 4: Please, add the descriptive statistics of each variable. We have adjusted Table 4 (now table 2)

Discussion

-BMJ Open, in the author guidelines, recommend writing short discussions. However, the discussed results are very poor with only four studies referenced. Others studies about ABCDE approach simulation training (Abelsson et al. Learning High-Energy Trauma Care Through Simulation. Clinical Simulation in Nursing 2018) or evaluation (Fernandez-Mendez et al. ABCDE approach to victims by lifeguards: How do they manage a critical patient? A cross sectional simulation study. PLoS One. 2019) could be discussed.

-The benefits and challenges of the simulation training should also be discussed. Some references that could help: Cook et al. Technology-enhanced simulation for health professions education: a

systematic review and meta-analysis JAMA 2011 / Grant et al. Difficult debriefing situations: A toolbox for simulation educators. Med Teach 2018).

Thank you for this suggestion and references to improve our discussion and limitations section. We have rewritten the discussion with more focus on the ABCDE (as reviewer 2 suggested), rewritten the limitations and added the references suggested.

Reviewer 2:

Reviewer Name: Christian Berger

Institution and Country: Charité Universitätsmedizin Berlin, Germany

- In the introduction, you describe the relevance of early resuscitation within the “golden hour”. Further you state, that the ABCDE approach is the primary assessment for trauma since decades. Please provide data regarding this statement and clarify the context of the two statements.

Thank you for this commentary, it helped us to make the introduction clearer. We have rewritten this part of the introduction and added more information about the use of ABCDE in trauma with references.

- In the methods, you mention three simulation scenarios for the measurements. Please present more information about the exact type and content of these scenarios.

Thank you for this suggestion to improve the methods sections. We have described the content of the scenario's in the methods / study design section.

- Where took they place, what kind of equipment and environment was provided for the participants. We have added more detailed information in the methods / intervention section and also in the methods / study setting and population section.

- How many people in which role participate in each scenario in total?

We have added more detailed information in the methods / intervention section and also in the methods / study setting and population section.

- Please revise the methodological section and state more clearly, which test was when more used. In this context, please make clearer, why the evaluation of the differences between time points was altered between separate skills (supplemental file).

Thank you for this commentary. Our apologies that our text created the impression that the evaluation of the differences between time points was altered between separate skills, whereas, actually, this evaluation was not altered. Your feedback comment stimulated us to reformulate the methods section, in order to avoid confusion, and to make clear that we used the Friedman's test for three related samples to analyse whether the total primary assessment scores of the entire group of participants differed between the three measurement moments. We used the Wilcoxon signed rank test for two related samples to analyse both total score on the primary assessment and separate skills at two different time points. We rewrote the data analysis section to explain this in more detail.

- The data collection took place at three different moments, separate from the course. When and how were they scheduled specifically and communicated with the participants.

We added more detailed information in the methods / study setting and population section.

- Were they able to prepare before data collection?

We had partially mentioned this in the discussion / limitations section, so we have rewritten this part to make it clearer.

- If observers were identical with the research team and furthermore involved as tutors in the course, please comment on potential bias.

The observers were not identical to the research team, so the videos could be (and were) offered “blinded” for the measurement moment. The observers and the research team consisted of course instructors. We have consciously involved course instructors in order to have observers who are experts in the ABCDE approach and to strive for uniformity in teaching and assessing the ABCDE. We have added a comment in the discussion / limitations section and in the methods / study protocol section.

- Your study was conducted within the courses between August 2012 and March 2014. How many physicians participate in your courses in total?

We added more detailed info at the results / characteristics of study subjects section: Between August 2012 and December 2013 twenty-seven courses were given to six participants each. From the total of 162 course participants thirty participants volunteered for this study.

- How was the number of 30 participants in your study selected? Did you perform a power analysis?

We didn't perform a power analysis, because we didn't know the expected effect. It was not possible to calculate a needed sample size. We decided to get a pilot sample of 30 because it seemed a realistic number to achieve in a year (but it took a few months longer). This relative small sample size already showed large significant differences.

In our statistical analysis we accounted for a small sample size by using the Wilcoxon signed-rank test. It is a non-parametric statistical hypothesis test used to compare two related samples to assess whether their population mean ranks differ. It can be used as an alternative to the paired Student's t-test when the sample size is small and the population cannot be assumed to be normally distributed.

- Has every one of the study participants fulfilled the data collection completely at all three time points?

All participants fulfilled all data collection at all three time points. The video recording of T3 of one participant was lost due to technical problems. This is described in the results / characteristics of study subjects section.

- Providing demographical data from your general course population would help the reader. Thank you for this comment. We added more detailed info at results / characteristics of study subjects section and information about the course participants in the methods / intervention section.

- You stated that ethics approval was not required, because intervention was not on behalf of the study due to course participation. What about the T3 measurement, which took place after the course? Please state this.

All the time measurements (T1, T2 and T3) were taken apart from the course and the course was the intervention. The volunteers would also have participated in the course if they wouldn't have participated in the study. A small amount of money was offered to compensate for the time and effort spend in the three measurements.

- Further, it was possible to not pass the course and it's not clear mentioned, if the observers were involved in data collection or participated as course instructors.

We added more detailed info in the methods / study protocol. The observers were not involved in the data collection, but they were course instructors.

- Who asked the participants to join the study and when?

There was only one researcher, who was also course instructor, who asked the course participants to participate in the study.

- Were observers or course instructors involved?

The observers were not involved in recruiting the study participants, only one course instructor approached the course participants.

- Asking for a “voluntarily” participation by potential future examiners may influence the participant’s decision. Please comment on this.

Thank you for this comment. The fact that a course instructor (and examiner) was involved in the recruitment could have been of influence on the participants decision. We strived for a save environment by a statement in the invitation e-mail that declining to participate in the study will not influence the course results.

We added a comment in the methods / study setting and population section.

- Please state, if the study is enlisted in registry e.g. “clingov”. If not so, this should be done.

We have critically read the instructions of Clingov, but because it was no clinical trial we think that this study does not meet the criteria to enlist it in Clingov.

- Some of the measured skills were not analyzed as they were scored too often as “does not apply”.

Please provide information about your cut off for analyzation and how it was calculated. Further, please provide the number of data sets in each category, which were finally analyzed.

Thank you for this suggestion to improve our results and tables. We have added this in the tables in the article and in the supplemental file for reviewers.

- The discussion sets a focus on communication and leadership. This can be summarized under the topic “CRM”. Regarding to your heading and introduction, I recommend focusing on the ABCDE approach or please make clear, how teaching your specific ABCDE course influences the CRM-abilities of the participants. Otherwise, I would suggest changing the topic of your paper.

We added more information in the methods / intervention section about how the course also teaches CRM-skills. We have rewritten the discussion with more focus on the ABCDE approach.

- You conclude that simulation training leads to the results of your measurement. Please clarify, why the simulation training and not for example the lecture of the book leads to the rise in the participants performance.

We think that the simulation training is the most contributing factor, because the course consists mainly of simulation training. There are only 2 lectures and 24 simulations. Although studying the book and following/attending lectures may, indeed, add to the learning effects, in general, the participants’ feedback comments after the course indicate that in particular performing so many simulations is useful. Moreover, previous research provided evidence that adding simulation training is associated with higher scores in an Emergency Medicine Clerkship than providing only lectures (Frallicciardi et al 2012, Cook et al 2019).

We have made some adjustments in the methods / intervention section and in the discussion / limitations section to address this issue in more detail.

- Further, is it possible to exclude, that training or experience outside the course leads to the improvement of participants behaviour during data collection? Was this evaluated? Pleas discuss this and accordingly adapt the conclusion.

We didn’t evaluate what factors outside the course might have influenced the participants performance. It would be interesting to investigate this in future research.

Some minor points:

- Please introduce the abbreviation ED in page 7 line 45 before use

- For data analysis you state a 5 point scale but using a 4 point scale with the additional option “not applicable”, please make this more clear

- Page 12 line 35, think you mean “lost” instead “last”

Thank you for these comments. We have adjusted these minor points.

VERSION 2 – REVIEW

REVIEWER	Cristian Abelairas-Gómez Universidade de Santiago de Compostela, Spain
REVIEW RETURNED	27-Sep-2019

GENERAL COMMENTS	General comments: - It is confusing the terminology “mean scores” and “mean rank scores”. It is not clear the difference between them. Both are used in the abstract and the reader will not be able to understand. In Data analysis authors state that “mean rank score is calculated by ranking the score of each participant on T1, T2 and T3”. Why is this measure important? The aim of the investigation is to study de effect of the simulation training in the learning and retaining of the ABCDE skills. Thus, the main data is the intra-group analysis among different test: T1 vs. T2 vs. T3. Specific comments: Abstract: - Replace “Thirty voluntary participants (mean age 27 years, 21 females, 9 males) of a simulation-based course” with “Thirty voluntary participants (21 females & 9 males; 27±2.77 years,) of a simulation-based course”.- Remove “(Table 1)” from the results section.- Those aspects regarding “mean scores” and “mean rank scores” should be emended. Introduction: - Authors state: “the importance of early treatment in the so-called 'golden hour' [...] has been recognized in several emergencies such as trauma, stroke, sepsis and shock [1-5, 8-11]”. However, some of the references used (1-5, 8-11) are related only to training without any mention of the named golden hour.- IGZ abbreviation is not necessary. It only appears once. Methods: - In the patient and public involvement section is required to add that participants signed a written informed consent, that they know all information about the investigation and that they could withdraw the study in any moment. All under the Helsinki declaration.- Replace “The measurements through video recordings were obtained before (T1), directly after (T2) and approximately three months after the intervention (T3).” with “The measurements through video recordings were obtained before (T1), directly after (T2) and 3-4 months after the intervention (T3).” This range of one month between T2 and T3 should be mentioned in limitations. One month is a third of the theoretical follow-up and is a bias that should be take into account in the results interpretation.- Intervention section should be expressed in past.
--

	- Primary assessment instrument is explained much better but I would like to ask two questions that I consider that should be mentioned: How was the scales 1-8, 1-7, 1-4 and 1-2 in the examinations of B, C, D and E respectively used in the score? “Does not apply” means that there are steps that were not required, but how was the evaluation if the participants performed a step that was not necessary? - Data analysis: Wilcoxon signed-rank test is suitable for non-parametric variables in an intra-group analysis. However, when there are more than one pair analysis (in this case there are three: T1 vs. T2 / T1 vs. T3 / T2 vs. T3) is more exact to perform a correction. One easy way might be to divide the significance level between the number of pairs: $0.05/3 = 0.017$. Results: - Clear definition of “mean scores” and “mean rank scores” should be added for a good understanding and interpretation of the results. In table 1 is not clear if Wilcoxon test refers to mean of mean rank score. - P14 – L45-55: The skills mentioned in this paragraph should be deleted since they are in table 2. - Table 2: Why is N=28 in two variables if only one video recording was lost? Discussion: - Reference [5]: In this manuscript the evaluation was performed in a simulation scenario. No specific data about the training (if simulation or not) was described. Limitations: - It was not taken into account the experience or previous training of the participants. This should be commented in this section. - The use of a sample of convenience without power analysis should be added in this section.
--	---

REVIEWER	Christian Berger Charite´ Universitätsmedizin Berlin, Campus Benjamin Franklin
REVIEW RETURNED	23-Oct-2019

GENERAL COMMENTS	To the author: The manuscript is revised properly with major changes according to the majority of the initial comments but some points have not yet been addressed and needs further attention. Your study is classified as a “observational intervention study”. In general, a study can be observational or interventional. Over all, it is a prospective design. Therefore, an ethic vote should be done. If this is not necessary due to your ethic regulations, please provide a comment from your ethical board. Please state, if the study is registered in a registry e.g. “clingov”. If
--

	not so, this should be done. Please discuss any potential underestimation of the 4-point rated items compared with 2-point rated items. Eg. due to a central tendency error it seems possible, that 4-point items may be underestimated. This may be aggravated due to an unbalanced distribution of 2 and 4-point ratings over all categories. Further, please provide in general the amount of “not-applicable” rated items. Why was 10 the cut-off? Was this crosschecked by a statistician? Please comment on this. If no crosscheck was performed, a review or comment by a statistician is recommend. During measurement, please comment on the role of the researcher participating as “non-obstructive nurse” and potential bias. How did the researchers prepare for their roles? How did you avoid general or researcher specific influence of your results during measurement? Was the amount and content of participant and researcher in “nurse-role” interaction measured and compared between the scenarios? Was the time for ABCDE approach measured? Did ABCDE evaluation take part in a comparable period over all measurements and scenarios? If possible, please present these data and comment. Providing comparisons of investigation-participants previous experience in emergency training and work experience with the other 132 course participants would help in rating your conclusion in general. Minor points: - please avoid change of scenario enumeration (I, II, III and A, B, C were used eg p. 8) - Providing an overview of the necessary education and experience level for being an instructor for this ABCDE-course would help the reader.
--	--

VERSION 2 – AUTHOR RESPONSE

Reviewer: 1

Reviewer Name: Cristian Abelairas-Gómez

Institution and Country: Universidade de Santiago de Compostela, Spain Please state any competing interests or state 'None declared': Non declared

General comments:

- It is confusing the terminology “mean scores” and “mean rank scores”. It is not clear the difference between them. Both are used in the abstract and the reader will not be able to understand. In Data analysis authors state that “mean rank score is calculated by ranking the score of each participant on T1, T2 and T3”. Why is this measure important? The aim of the investigation is to study de effect of the simulation training in the learning and retaining of the ABCDE skills. Thus, the main data is the intra-group analysis among different test: T1 vs. T2 vs. T3.

Thank you for this comment. We agree that adding mean scores to mean rank scores might cause confusion. Therefore, we are glad to remove the mean scores from the manuscript.

With regard to statistics: we have chosen non-parametric analysis of data because of skewed data distribution. After consulting a statistical expert, we used the Wilcoxon signed rank and Friedman test to analyse our research question as the Wilcoxon signed-rank test aims to detect differences between variables from the same sample before and after an intervention by calculating the differences between their ranks.

Specific comments:

Abstract:

- Replace “Thirty voluntary participants (mean age 27 years, 21 females, 9 males) of a simulation-based course” with “Thirty voluntary participants (21 females & 9 males; 27 ± 2.77 years,) of a simulation-based course”. We have replaced this text.

- Remove “(Table 1)” from the results section. We have removed this text.

- Those aspects regarding “mean scores” and “mean rank scores” should be emended. We have removed the mean scores in text and table.

Introduction:

- Authors state: “the importance of early treatment in the so-called 'golden hour' [...] has been recognized in several emergencies such as trauma, stroke, sepsis and shock [1-5, 8-11]”. However, some of the references used (1-5, 8-11) are related only to training without any mention of the named golden hour.

Thank you for noting this. We meant all references refer to the importance of early treatment and a few all so refer to golden hour. We have adapted this by removing “the golden hour”.

- IGZ abbreviation is not necessary. It only appears once. We have removed the abbreviation.

Methods:

- In the patient and public involvement section is required to add that participants signed a written informed consent, that they know all information about the investigation and that they could withdraw the study in any moment. All under the Helsinki declaration.

Ethical approval was waived by our medical ethics committee (METc UMC Groningen) as this research is educational research. We have followed our institutional research guideline (UMCG kaderreglement nWMO, versie 2.0, mei 2017) for educational research. Participants provided verbal consent.

We have added this information in the patient and public involvement section.

- Replace “The measurements through video recordings were obtained before (T1), directly after (T2) and approximately three months after the intervention (T3).” with “The measurements through video recordings were obtained before (T1), directly after (T2) and 3-4 months after the intervention (T3).” This range of one month between T2 and T3 should be mentioned in limitations. One month is a third of the theoretical follow-up and is a bias that should be take into account in the results interpretation. We have changed the text in the method section and added a remark in the limitations section.

- Intervention section should be expressed in past. We have adapted this.

- Primary assessment instrument is explained much better but I would like to ask two questions that I consider that should be mentioned: How was the scales 1-8, 1-7, 1-4 and 1-2 in the examinations of B, C, D and E respectively used in the score?

The lowest score was 0, the highest score was 1. For example in the B there was a maximum of 8 items to examine during physical examination. If one item was examined the score was $1/8 = 0.125$, if two items were examined the score was $2/8 = 0.25$, if three items were examined, the score was $3/8 = 0.375$, etc. So, the highest possible score on complete examination in the B was $8/8 = 1$.

We have added this explanation in the measurements section.

“Does not apply” means that there are steps that were not required, but how was the evaluation if the participants performed a step that was not necessary?

This was not specifically registered, but depending on what steps were taken unnecessary this could be scored under de remaining items as less self-confident or less adequate clinical reasoning.

- Data analysis: Wilcoxon signed-rank test is suitable for non-parametric variables in an intra-group analysis. However, when there are more than one pair analysis (in this case there are three: T1 vs. T2 / T1 vs. T3 / T2 vs. T3) it is more exact to perform a correction. One easy way might be to divide the significance level between the number of pairs: $0.05/3 = 0.017$.

We have now applied the Holm correction which can be used to counteract the problem of multiple comparisons. That changed two of the separate skills in non-significant difference, so we have adapted this in the text and in the supplemental file.

Results:

- Clear definition of “mean scores” and “mean rank scores” should be added for a good understanding and interpretation of the results. In table 1 it is not clear if Wilcoxon test refers to mean of mean rank score. We have removed the mean scores in text and table.

- P14 – L45-55: The skills mentioned in this paragraph should be deleted since they are in table 2. We have adapted this.

- Table 2: Why is N=28 in two variables if only one video recording was lost?
One observer had not scored these two items in one scenario of one participant by accident.

Discussion:

- Reference [5]: In this manuscript the evaluation was performed in a simulation scenario. No specific data about the training (if simulation or not) was described.

Thank you for noting this. We have adapted this.

Limitations:

- It was not taken into account the experience or previous training of the participants. This should be commented in this section. We have added a comment in the limitations section.

- The use of a sample of convenience without power analysis should be added in this section. We have added a comment in the limitations section.

Reviewer: 2

Reviewer Name: Christian Berger

Institution and Country: Charité Universitätsmedizin Berlin, Campus Benjamin Franklin Please state any competing interests or state ‘None declared’: None declared

Please leave your comments for the authors below To the author:

The manuscript is revised properly with major changes according to the majority of the initial comments but some points have not yet been addressed and needs further attention.

-Your study is classified as a “observational intervention study”. In general, a study can be observational or interventional. Overall, it is a prospective design. Therefore, an ethics vote should be done. If this is not necessary due to your ethics regulations, please provide a comment from your ethical board.

Our study is observational educational research. Ethical approval was waived by our medical ethics committee (METc UMC Groningen) as this research is educational research. We have added an independent review board declaration.

-Please state, if the study is registered in a registry e.g. “clinicaltrials.gov”. If not so, this should be done. Our study is not a clinical trial, but educational research. In our study “the intervention” is an already existing course. So, we analyzed normal practice.

We have followed our institutional research guideline (UMCG kaderreglement nWMO, versie 2.0, mei 2017) for educational research and registered the study in the research register from our hospital. We have critically read the instructions of Clingov, but because it was no clinical trial that assesses biomedical or health outcomes we think that this study does not meet the criteria to enlist it in Clingov. This is stated in the instruction of Clingov: "ClinicalTrials.gov allows the registration of clinical studies with human subjects that assess biomedical and/or health outcomes"

-Please discuss any potential underestimation of the 4-point rated items compared with 2-point rated items. Eg. due to a central tendency error it seems possible, that 4-point items may be underestimated. This may be aggravated due to an unbalanced distribution of 2 and 4-point ratings over all categories. A statistician recommended to allocate a maximum score of 1 to each item, so each item weighs equally. The unbalanced distribution of 4 and 2 point ratings may only influence the score of the total primary assessment, but this is the case at all three measurements. The unbalanced distribution of 4 and 2 point ratings will not influence the score and the analysis on the separate skills.

-Further, please provide in general the amount of "not-applicable" rated items. The amount of not applicable rated items was in ten items between 0-3, in five items between 3-10, in two items between 10-20 and in four items > 20. We have added this information in the limitations section.

-Why was 10 the cut-off? Was this crosschecked by a statistician? Please comment on this. If no crosscheck was performed, a review or comment by a statistician is recommended. It was not possible to perform the Wilcoxon signed rank test in SPSS when $N < 10$.

-During measurement, please comment on the role of the researcher participating as "non-obstructive nurse" and potential bias. How did the researchers prepare for their roles? How did you avoid general or researcher specific influence of your results during measurement? In our study we have deliberately chosen for a researcher participating as "non-obstructive nurse" in the measurement to minimize potential bias caused by help from the "non-obstructive nurse". The researcher knew the research questions and were instructed in detail to only follow instructions from the participant and not help in any way. We did not schedule the researchers and operators with an equal distribution over the measurement moments, but all five researchers rotated between roles of the nurse and operator on own initiative. We think the bias of the nurse influencing the participant is negligible. We have added this comment in the limitation section.

-Was the amount and content of participant and researcher in "nurse-role" interaction measured and compared between the scenarios? No it was not measured.

-Was the time for ABCDE approach measured? Did ABCDE evaluation take part in a comparable period over all measurements and scenarios? If possible, please present these data and comment. No it was not measured.

-Providing comparisons of investigation-participants previous experience in emergency training and work experience with the other 132 course participants would help in rating your conclusion in general. We have no information over work experience and training experience of the other 132 course participants, so we are not able to make this comparison.

Minor points:

- please avoid change of scenario enumeration (I, II, III and A, B, C were used eg p. 8)

Thank you for noting this inconsistency. We have adapted this.

- Providing an overview of the necessary education and experience level for being an instructor for this ABCDE-course would help the reader.

Each instructor for this course has to follow a formalized educational program to become an instructor: First they have to pass the course as participant and have to work in the field of emergency medicine or acute care. Second they need to follow a two-day generic instructor course specifically developed for simulation training. Then they have to act as assistant-trainer for at least two courses and they need to write a report reflecting on their own role as instructor. Finally they are observed by an experienced instructor to become certified. As instructor, they have to teach the course least twice a year to stay competent and they need to follow the course-specific instructors day each year. We have added this information in the intervention section of the manuscript.

VERSION 3 - REVIEW

REVIEWER	Cristian Abelairas-Gómez Universidade de Santiago de Compostela
REVIEW RETURNED	14-Jan-2020

GENERAL COMMENTS	Authors made a great work attending the major comments from the second revision. Now, I only have some minor suggestions. Please, pay attention in the comment* about the mean score and the mean rank score. Either I did not understand well, or mean score was removed instead of mean rank score:  - Measurements: The explanation to calculate the score in each part of the ABCDE approach should be added in the manuscript. Please, replace "Because some skills or competences were marked as not applicable, we calculated mean scores in each category (A, B, C, D, E and remaining items) based on the skills and competences which actually were applicable. In each category the maximal score to obtain was 1. Therefore, the maximal total score to obtain on the primary assessment for each scenario was 6 and the minimal score was 0." with "Because some skills or competences were marked as not applicable, we calculated mean scores in each category (A, B, C, D, E and remaining items) based on the skills and competences which actually were applicable. In each category the maximal score to obtain was 1. For example in the B there was 8 items to examine. If one item was examined the score was $1/8 = 0.125$, if two items were examined the score was $2/8 = 0.25$, etc. Thus, the highest possible score on complete examination in the B was $8/8 = 1$. The maximal total score to obtain on the primary assessment for each scenario was 6 and the minimal score was 0." - Data analysis:  o Please, report which is the inter-observer reliability rate considered at least "Acceptable" for the Spearman rank. o Please, replace "The Friedman test compares the mean rank scores at T1, T2 and T3. The mean rank score is calculated by ranking the score of each participant on T1, T2 and T3 and then calculating the mean rank of the entire group on T1, T2 and T3" with "Mean rank score was calculated in each test by ranking the score of each participant. Friedman test was used to compare the mean rank scores at T1, T2 and T3." - Results:  o Authors write about a total of 41 skills or competences. However, Figure 1 shows 40.
--

	o Figure 2 is not mentioned in the text. May you provide the mean as well? Please, delete the “89” from the figure and replace “1,00”, “2,00” and “3,00” with “T1” “T2” and “T3”. o *How is it possible the huge difference between the mean rank score and the median? In all test the mean rank score is lower than Q1. The results of the mean rank described in the another version of the manuscript (2.90, 5.06 & 4.67) have more sense than the scores provided in the last version (1.14, 2.62 & 2.24). In another revision of the manuscript, authors were asked about the mean score and the mean rank score. Understanding the confusion caused by both scores, authors decided to remove the mean score. However, attending to the data shown in Figure 2 (median, Q1 & Q3) and the aim of the study, maybe mean rank score should have been removed instead of mean score. In addition, in “Data analysis” section, authors talk again of both mean rank score and mean score. The most important data for the reader in this regard, are the mean marks in T1 vs. mean marks in T2 vs. mean marks in T3. And this should be written very clear.
--	---

VERSION 3 – AUTHOR RESPONSE

- Measurements: The explanation to calculate the score in each part of the ABCDE approach should be added in the manuscript.

Please, replace “Because some skills or competences were marked as not applicable, we calculated mean scores in each category (A, B, C, D, E and remaining items) based on the skills and competences which actually were applicable. In each category the maximal score to obtain was 1. Therefore, the maximal total score to obtain on the primary assessment for each scenario was 6 and the minimal score was 0.” with “Because some skills or competences were marked as not applicable, we calculated mean scores in each category (A, B, C, D, E and remaining items) based on the skills and competences which actually were applicable. In each category the maximal score to obtain was 1. For example in the B there was 8 items to examine. If one item was examined the score was $1/8 = 0.125$, if two items were examined the score was $2/8 = 0.25$, etc. Thus, the highest possible score on complete examination in the B was $8/8 = 1$. The maximal total score to obtain on the primary assessment for each scenario was 6 and the minimal score was 0.”

Thank you for this remark. We have added this part in the previous version in the track changes document, but accidentally deleted it in the document without track changes. We have put it back in the text were it is applicable.

- Data analysis:

o Please, report which is the inter-observer reliability rate considered at least “Acceptable” for the Spearman rank.

A correlation coefficient lower than 0.5 is considered as weak correlation, a correlation coefficient between 0.5 and 0.7 is considered as moderate correlation, a correlation coefficient between 0.7 and 0.9 is considered as high correlation and a correlation coefficient between 0.9 and 1 is considered as very high correlation.

We have added this in the text.

o Please, replace “The Friedman test compares the mean rank scores at T1, T2 and T3. The mean rank score is calculated by ranking the score of each participant on T1, T2 and T3 and then calculating the mean rank of the entire group on T1, T2 and T3” with “Mean rank score was calculated in each test by ranking the score of each participant. Friedman test was used to compare the mean rank scores at T1, T2 and T3.”

Thank you for this commentary. It showed us that from our text it seems that the calculation of the mean rank is not part of the Friedman (and Wilcoxon) and that we did the calculation ourselves. But both tests calculate and compare the mean ranks. We added the word “calculate” to make it more clear and removed the word “scores” after mean rank because it refers to ranks of the measurement moments and not to the achieved scores on the primary assessment or a skill/competence.

- Results:

o Authors write about a total of 41 skills or competences. However, Figure 1 shows 40.

Thank you for noticing this inconsistency. We have adapted this number in the text.

o Figure 2 is not mentioned in the text. May you provide the mean as well? Please, delete the “89” from the figure and replace “1,00”, “2,00” and “3,00” with “T1” “T2” and “T3”.

In the boxplot we used means because we were not able to make a boxplot with mean ranks. We used the boxplot to make the differences between the scores on the different measurement moments more visual.

But because adding Figure 2 with boxplots (and mentioning the mean scores) causes confusion about the statistical approach we consulted a statistician again. He insured us that our statistical approach (Friedman and Wilcoxon) was right and advised us to replace the boxplots with a line graph which shows the trend in time of each individual participant.

So, we have changed Figure 2.

o *How is it possible the huge difference between the mean rank score and the median?

The Median is on a scale from 0 to 6, because this was the maximal total score to obtain on each scenario and therefore on each measurement. The mean rank is calculated on a scale from 1-3, because when three items are ranked the best rank is 1 and the worst rank is 3.

In all tests the mean rank score is lower than Q1. The results of the mean rank described in the another version of the manuscript (2.90, 5.06 & 4.67) have more sense than the scores provided in the last version (1.14, 2.62 & 2.24).

We have looked back in the previous versions of the manuscript, but as far as we know we never mentioned the mean rank as (2.90, 5.06 & 4.67). We only used these numbers as mean scores on the total primary assessment of the whole group at the different measurement moments. The mean rank has always been described as 1.14, 2.62 and 2.24.

In another revision of the manuscript, authors were asked about the mean score and the mean rank score. Understanding the confusion caused by both scores, authors decided to remove the mean

score. However, attending to the data shown in Figure 2 (median, Q1 & Q3) and the aim of the study, maybe mean rank score should have been removed instead of mean score.

We would like to stick to the mean rank as argued in our previous response.

After consulting a statistical expert, we used the non-parametric Wilcoxon signed rank and Friedman test for skewed data distribution, to analyse our research question as the Wilcoxon signed-rank test aims to detect differences between variables from the same sample before and after an intervention by calculating the differences between their ranks.

To make this more clear we delete the word “score” after mean rank, because it refers to ranks of de measurement moments and not to the achieved scores. To avoid further confusion we also replaced Figure 2 as mentioned before.

In addition, in “Data analysis” section, authors talk again of both mean rank score and mean score. We mentioned the mean score in the analysis of inter-observer reliability not in the analysis of our research question. We meant that the scores of both observers could be averaged to do further analysis with one mean score instead of two score from both observers.

We have adapted the text to avoid confusion.

We also mention mean scores in the “measurements” part of the manuscript. In this part we describe how we handled missing values. These mean scores mentioned here are used to calculate a total score for each individual participant.

The most important data for the reader in this regard, are the mean marks in T1 vs. mean marks in T2 vs. mean marks in T3. And this should be written very clear.

We fully agree that the results should be written very clear. Therefore we thank the reviewer for the comments and have we adapted the manuscript as described before. We hope that these adjustments make it more clear.

VERSION 4 - REVIEW

REVIEWER	Cristian Abelairas-Gómez Universidade de Santiago de Compostela (Spain)
REVIEW RETURNED	11-Feb-2020

GENERAL COMMENTS	How can the reader understand how this 1-3 scale was calculated? Much more clarification is needed. Each scale used must be explained. Means has to be described with standard deviation, and median with interquartile range. Why was figure 2 changed? The new one is an amount of lines that describes worse the results than the firts version. Please, change again and include the suggestions of the last revision of the manuscript in this regard.
--

VERSION 4 – AUTHOR RESPONSE

1. How can the reader understand how this 1-3 scale was calculated? Much more clarification is needed. Each scale used must be explained.

The mean rank is calculated on a scale from 1-3, because when three items (three measurement moments) are ranked, the best rank is 1 and the worst rank is 3.

We have added this in the text.

2. Means has to be described with standard deviation, and median with interquartile range.

We have added the interquartile range of the median in the table, but we did not add means and standard deviation, because in the previous revisions we removed the means as reviewer Abelairas-Gómez showed us that using both mean and mean rank would cause confusion.

We consulted our statistician again and he confirmed that using Friedman and Wilcoxon (analysis with mean ranks) for non-parametric paired measurements is the correct approach. We already explained in our previous response to the reviewer that – based on the advice of our expert statistician – we removed the means.

3. Why was figure 2 changed? The new one is an amount of lines that describes worse the results than the first version. Please, change again and include the suggestions of the last revision of the manuscript in this regard.

We changed the figure back in the boxplot showing median and interquartile range, but in view of the advice of an expert in statistics, see our previous response, we decided to avoid means and to be consequent by using mean ranks, median and total scores as well in tables, text and figures.

We did remove the “89” outlier as suggested in the previous comments of the reviewer.